# ABSTRACTIVE DIALOG SUMMARIZATION
# WITH SEMANTIC SCAFFOLDS

## ABSTRACT

The demand for abstractive dialog summary is growing in real-world applications. For example, customer service center or hospitals would like to summarize customer service interaction and doctor-patient interaction. However, few researchers explored abstractive summarization on dialogs due to the lack of suitable datasets. We propose an abstractive dialog summarization dataset based on MultiWOZ (Budzianowski et al., 2018). If we directly apply previous state-of-the-art document summarization methods on dialogs, there are two significant drawbacks: the informative entities such as restaurant names are difficult to preserve, and the contents from different dialog domains are sometimes mismatched. To address these two drawbacks, we propose Scaffold Pointer Network (SPNet) to utilize the existing annotation on speaker role, semantic slot and dialog domain. SPNet incorporates these semantic scaffolds for dialog summarization. Since ROUGE cannot capture the two drawbacks mentioned, we also propose a new evaluation metric that considers critical informative entities in the text. On MultiWOZ, our proposed SPNet outperforms state-of-the-art abstractive summarization methods on all the automatic and human evaluation metrics.

## 1 INTRODUCTION

Summarization aims to condense a piece of text to a shorter version, retaining the critical information. On dialogs, summarization has various promising applications in the real world. For instance, the automatic doctor-patient interaction summary can save doctors' massive amount of time used for filling medical records. There is also a general demand for summarizing meetings in order to track project progress in the industry. Generally, multi-party conversations with interactive communication are more difficult to summarize than single-speaker documents. Hence, dialog summarization will be a potential field in summarization track.

There are two types of summarization: extractive and abstractive. Extractive summarization selects sentences or phrases directly from the source text and merges them to a summary, while abstractive summarization attempts to generate novel expressions to condense information. Previous dialog summarization research mostly study extractive summarization (Murray et al., 2005; Maskey & Hirschberg, 2005). Extractive methods merge selected important utterances from a dialog to form summary. Because dialogs are highly dependant on their histories, it is difficult to produce coherent discourses with a set of non-consecutive conversation turns. Therefore, extractive summarization is not the best approach to summarize dialogs. However, most modern abstractive methods focus on single-speaker documents rather than dialogs due to the lack of dialog summarization corpora. Popular abstractive summarization dataset like CNN/Daily Mail (Hermann et al., 2015) is on news documents. AMI meeting corpus (McCowan et al., 2005) is the common benchmark, but it only has extractive summary.

In this work, we introduce a dataset for abstractive dialog summarization based on MultiWOZ (Budzianowski et al., 2018). Seq2Seq models such as Pointer-Generator (See et al., 2017) have achieved high-quality summaries of news document. However, directly applying a news summarizer to dialog results in two drawbacks: informative entities such as place name are difficult to capture precisely and contents in different domains are summarized unequally. To address these problems, we propose Scaffold Pointer Network (SPNet). SPNet incorporates three types of semantic scaffolds in dialog: speaker role, semantic slot, and dialog domain. Firstly, SPNet adapts separate encoder

to attentional Seq2Seq framework, producing distinct semantic representations for different speaker roles. Then, our method inputs delexicalized utterances for producing delexicalized summary, and fills in slot values to generate complete summary. Finally, we incorporate dialog domain scaffold by jointly optimizing dialog domain classification task along with the summarization task. We evaluate SPNet with both automatic and human evaluation metrics on MultiWOZ. SPNet outperforms Pointer-Generator (See et al., 2017) and Transformer (Vaswani et al., 2017) on all the metrics.

## 2 RELATED WORK

Rush et al. (2015) first applied modern neural models to abstractive summarization. Their approach is based on Seq2Seq framework (Sutskever et al., 2014) and attention mechanism (Bahdanau et al., 2015), achieving state-of-the-art results on Gigaword and DUC-2004 dataset. Gu et al. (2016) proposed copy mechanism in summarization, demonstrating its effectiveness by combining the advantages of extractive and abstractive approach. See et al. (2017) applied pointing (Vinyals et al., 2015) as copy mechanism and use coverage mechanism (Tu et al., 2016) to discourage repetition. Most recently, reinforcement learning (RL) has been employed in abstractive summarization. RL-based approaches directly optimize the objectives of summarization (Ranzato et al., 2016; Celikyilmaz et al., 2018). However, deep reinforcement learning approaches are difficult to train and more prone to exposure bias (Bahdanau et al., 2017).

Recently, pre-training methods are popular in NLP applications. BERT (Devlin et al., 2018) and GPT (Radford et al., 2018) have achieved state-of-the-art performance in many tasks, including summarization. For instance, Zhang et al. (2019) proposed a method to pre-train hierarchical document encoder for extractive summarization. Hoang et al. (2019) proposed two strategies to incorporate a pre-trained model (GPT) to perform the abstractive summarizer and achieved a better performance. However, there has not been much research on adapting pre-trained models to dialog summarization.

Dialog summarization, specifically meeting summarization, has been studied extensively. Previous work generally focused on statistical machine learning methods in extractive dialog summarization: Galley (2006) used skip-chain conditional random fields (CRFs) (Lafferty et al., 2001) as a ranking method in extractive meeting summarization. Wang & Cardie (2013) compared support vector machines (SVMs) (Cortes & Vapnik, 1995) with LDA-based topic models (Blei et al., 2003) for producing decision summaries. However, abstractive dialog summarization was less explored due to the lack of a suitable benchmark. Recent work (Wang & Cardie, 2016; Goo & Chen, 2018; Pan et al., 2018) created abstractive dialog summary benchmarks with existing dialog corpus. Goo & Chen (2018) annotated topic descriptions in AMI meeting corpus as the summary. However, topics they defined are coarse, such as "industrial designer presentation". They also proposed a model with a sentence-gated mechanism incorporating dialog acts to perform abstractive summarization. Moreover, Li et al. (2019) first built a model to summarize audio-visual meeting data with an abstractive method. However, previous work has not investigated the utilization of semantic patterns in dialog, so we explore it in-depth in our work.

## 3 PROPOSED METHOD

As discussed above, state-of-the-art document summarizers are not applicable in conversation settings. We propose Scaffold Pointer Network (SPNet) based on Pointer-Generator (See et al., 2017). SPNet incorporates three types of semantic scaffolds to improve abstractive dialog summarization: speaker role, semantic slot and dialog domain.

### 3.1 BACKGROUND

We first introduce Pointer-Generator (See et al., 2017). It is a hybrid model of the typical Seq2Seq attention model (Nallapati et al., 2016) and pointer network (Vinyals et al., 2015). Seq2Seq framework encodes source sequence and generates the target sequence with the decoder. The input sequence is fed into the encoder token by token, producing the encoder hidden states $h_i$ in each encoding step. The decoder receives word embedding of the previous word and generates a distribution to decide the target element in this step, retaining decoder hidden states $s_t$. In Pointer-Generator, attention

distribution $a^t$ is computed as in Bahdanau et al. (2015):

$$e_i^t = v^T \tanh\left(W_h h_i + W_s s_t + b_{\text{attn}}\right)$$
$$a^t = \text{softmax}\left(e^t\right) \tag{1}$$

where $W_h$, $W_s$, $v$ and $b_{attn}$ are all learnable parameters.

With the attention distribution $a^t$, context vector $h_t^*$ is computed as the weighted sum of encoder's hidden states. Context vector is regarded as the attentional information in the source text:

$$h_t^* = \sum_i a_i^t h_i \tag{2}$$

Pointer-Generator differs from typical Seq2Seq attention model in the generation process. The pointing mechanism combines copying words directly from the source text with generating words from a fixed vocabulary. Generation probability $p_{gen}$ is calculated as "a soft switch" to choose from copy and generation:

$$p_{\text{gen}} = \sigma\left(w_{h^*}^T h_t^* + w_s^T s_t + w_x^T x_t + b_{\text{ptr}}\right) \tag{3}$$

where $x_t$ is the decoder input, $w_{h^*}$, $w_s$, $w_x$ and $b_{ptr}$ are all learnable parameters. $\sigma$ is sigmoid function, so the generation probability $p_{gen}$ has a range of $[0, 1]$.

The ability to select from copy and generation corresponds to a dynamic vocabulary. Pointer network forms an extended vocabulary for the copied tokens, including all the out-of-vocabulary(OOV) words appeared in the source text. The final probability distribution $P(w)$ on extended vocabulary is computed as follows:

$$P_{\text{vocab}} = \text{softmax}\left(V'\left(V\left[s_t, h_t^*\right] + b\right) + b'\right)$$
$$P(w) = p_{\text{gen}} P_{\text{vocab}}(w) + (1 - p_{\text{gen}}) \sum_{i:w_i=w} a_i^t \tag{4}$$

where $P_{vocab}$ is the distribution on the original vocabulary, $V'$, $V$, $b$ and $b'$ are learnable parameters used to calculate such distribution.

## 3.2 SCAFFOLD POINTER NETWORK (SPNET)

Our Scaffold Pointer Network (depicted in Figure 1) is based on Pointer-Generator (See et al., 2017). The contribution of SPNet is three-fold: separate encoding for different roles, incorporating semantic slot scaffold and dialog domain scaffold.

### 3.2.1 SPEAKER ROLE SCAFFOLD

Our encoder-decoder framework employs separate encoding for different speakers in the dialog. User utterances $x_t^{usr}$ and system utterances $x_t^{sys}$ are fed into a user encoder and a system encoder separately to obtain encoder hidden states $h_i^{usr}$ and $h_i^{sys}$. The attention distributions and context vectors are calculated as described in section 3.1. In order to merge these two encoders in our framework, the decoder's hidden state $s_0$ is initialized as:

$$s_0 = concat(h_T^{usr}, h_T^{sys}) \tag{5}$$

The pointing mechanism in our model follows the Equation 3, and we obtain the context vector $h_t^*$:

$$h_t^* = concat(\sum_i a_i^{usr_t} h_i^{usr}, \sum_i a_i^{sys_t} h_i^{sys}) \tag{6}$$

### 3.2.2 SEMANTIC SLOT SCAFFOLD

We integrate semantic slot scaffold by performing delexicalization on original dialogs. Delexicalization is a common pre-processing step in dialog modeling. Specifically, delexicalization replaces the slot values with its semantic slot name(e.g. replace 18:00 with [time]). It is easier for the language modeling to process delexicalized texts, as they have a reduced vocabulary size. But these generated sentences lack the semantic information due to the delexicalization. Some previous dialog system

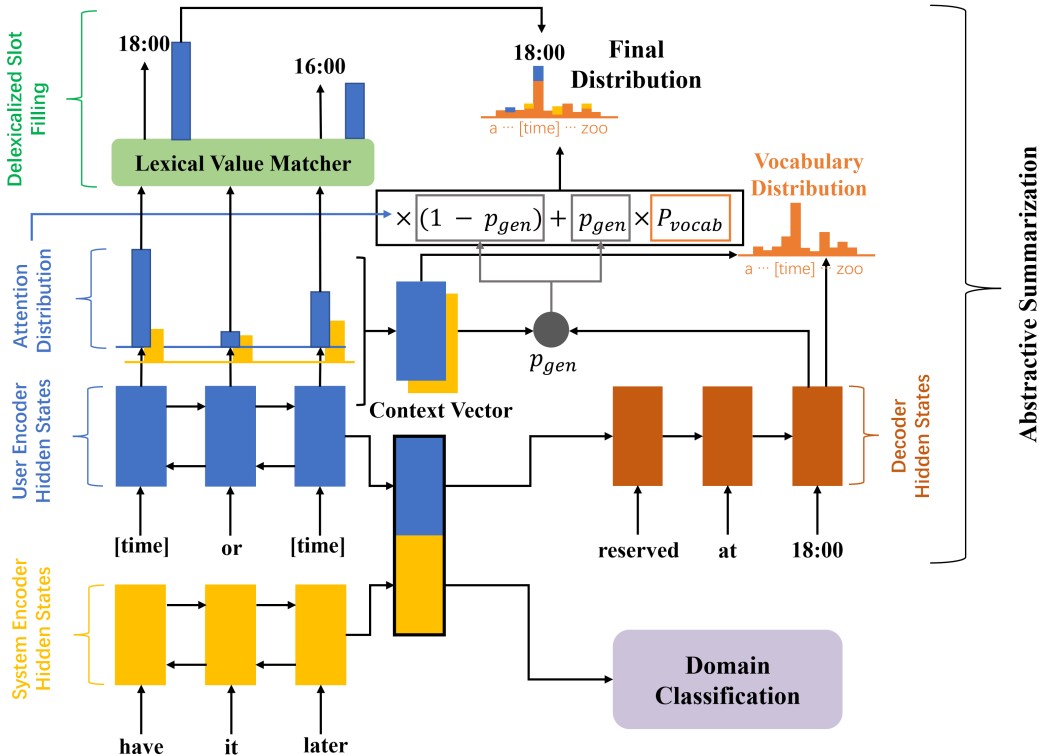

Figure 1: SPNet overview. The blue and yellow box is the user and system encoder respectively. The encoders take the delexicalized conversation as input. The slots values are aligned with their slots position. Pointing mechanism merges attention distribution and vocabulary distribution to obtain the final distribution. We then fill the slots values into the slot tokens to convert the template to a complete summary. SPNet also performs domain classification to improve encoder representation.

research ignored this issue (Wen et al., 2015) or completed single delexicalized utterance (Sharma et al., 2017) as generated response. We propose to perform delexicalization in dialog summary, since delexicalized utterances can simplify dialog modeling. We fill the generated templates with slots with the copy and pointing mechanism.

We first train the model with the delexicalized utterance. Attention distribution $a^t$ over the source tokens instructs the decoder to fill up the slots with lexicalized values:

$$value(w_{slot}) = \max_{\substack{value(w_i): \\ slot(w_i)=w_{slot}}} a_i^t \tag{7}$$

Note that $w_{slot}$ specifies the tokens that represents the slot name (e.g. [hotel_place], [time]). Decoder directly copies lexicalized value $value(w_i)$ conditioned on attention distribution $a_i^t$. If $w$ is not a slot token, then the probability $P(w)$ is calculated as Equation 4.

### 3.2.3 DIALOG DOMAIN SCAFFOLD

We integrate dialog domain scaffold through a multi-task framework. Dialog domain indicates different conversation task content, for example, booking hotel, restaurant and taxi in MultiWOZ dataset. Generally, the content in different domains varies so multi-domain task summarization is more difficult than single-domain. We include domain classification as the auxiliary task to incorporate the prior that different domains have different content. Feedback from the domain classification task provides domain specific information for the encoder to learn better representations. For domain classification, we feed the concatenated encoder hidden state through a binary classifier with two linear layers, producing domain probability $d$. The $i^{th}$ element $d_i$ in $d$ represents the probability

of the $i^{th}$ domain:

$$d = \sigma(U'(ReLU(U[h_T^{usr}, h_T^{sys}] + b_d)) + b_d') \tag{8}$$

where $U$, $U'$, $b_d$ and $b_d'$ are all trainable parameters in the classifier. We denote the loss function of summarization as $loss_1$ and domain classification as $loss_2$. Assume target word at timestep $t$ is $w_t^*$, $loss_1$ is the arithmetic mean of the negative log likelihood of $w_t^*$ over the generated sequence:

$$loss_1 = \frac{1}{T} \sum_{t=0}^{T} - \log P(w_t^*) \tag{9}$$

The domain classification task is a multi-label binary classification problem. We use binary cross entropy loss between the $i^{th}$ domain label $\hat{d}_i$ and predict probability $d_i$ for this task:

$$loss_2 = \frac{1}{|D|} \sum_{i=1}^{|D|} \hat{d}_i \log d_i + (1 - \hat{d}_i) \log (1 - d_i) \tag{10}$$

where $|D|$ is the number of domains. Finally, we reweight the classification loss with hyperparameter $\lambda$ and the objective function is:

$$loss = loss_1 + \lambda loss_2 \tag{11}$$

## 4 EXPERIMENTAL SETTINGS

### 4.1 DATASET

We validate SPNet on MultiWOZ-2.0 dataset (Budzianowski et al., 2018). MultiWOZ consists of multi-domain conversations between a tourist and a information center clerk on varies booking tasks or domains, such as booking restaurants, hotels, taxis, etc. There are 10,438 dialogs, spanning over seven domains. 3,406 of them are single-domain (8.93 turns on average) and 7,302 are multi-domain (15.39 turns on average). During MultiWOZ data collection, instruction is provided for crowd workers to perform the task. We use the instructions as the dialog summary, and an example data is shown in Table 2. Dialog domain label is extracted from existing MultiWOZ annotation. In the experiment, we split the dataset into 8,438 training, 1,000 validation, and 1,000 testing.

### 4.2 EVALUATION METRICS

ROUGE (Lin, 2004) is a standard metric for summarization, designed to measure the surface word alignment between a generated summary and a human written summary. We evaluate our model with ROUGE-1, ROUGE-2 and ROUGE-L. They measure the word-overlap, bigram-overlap, and longest common sequence between the reference summary and the generated summary respectively. We obtain ROUGE scores using the files2rouge package[1]. However, ROUGE is insufficient to measure summarization performance. The following example shows its limitations:

Reference: You are going to [restaurant_name] at [time].
Summary: You are going to [restaurant_name] at.

In this case, the summary has a high ROUGE score, as it has a considerable proportion of word overlap with the reference summary. However, it still has poor relevance and readability, for leaving out one of the most critical information: [time]. ROUGE treats each word equally in computing n-gram overlap while the informativeness actually varies: common words or phrases (e.g. "You are going to") significantly contribute to the ROUGE score and readability, but they are almost irrelevant to essential contents. The semantic slot values (e.g. [restaurant_name], [time]) are more essential compared to other words in the summary. However, ROUGE did not take this into consideration. To address this drawback in ROUGE, we propose a new evaluation metric: Critical Information Completeness (CIC). Formally, CIC is a recall of semantic slot information between a candidate summary and a reference summary. CIC is defined as follows:

$$CIC = \frac{\sum_{v \in V} Count_{match}(v)}{m} \tag{12}$$

---

[1] https://github.com/pltrdy/files2rouge

| Models | ROUGE-1 | ROUGE-2 | ROUGE-L | CIC |
|---|---|---|---|---|
| base (Pointer-Gen) (See et al., 2017) | 62.89 | 48.61 | 59.30 | 42.47 |
| Transformer (Vaswani et al., 2017) | 63.12 | 50.63 | 61.04 | 42.84 |
| base + speaker role | 72.01 | 60.55 | 68.40 | 53.08 |
| base + speaker role + semantic slot | 90.68 | 83.54 | 84.36 | 70.25 |
| SPNet (base + speaker role + semantic slot + dialog domain) | **90.97** | **84.14** | **85.00** | **70.45** |

Table 1: Automatic evaluation results on MultiWOZ. We use Pointer-Generator as the base model and gradually add different semantic scaffolds.

where $V$ stands for a set of delexicalized values in the reference summary, $Count_{match}(v)$ is the number of values co-occurring in the candidate summary and reference summary, and $m$ is the number of values in set $V$. In our experiments, CIC is computed as the arithmetic mean over all the dialog domains to retain the overall performance.

CIC is a suitable complementary metric to ROUGE because it accounts for the most important information within each dialog domain. CIC can be applied to any summarization task with predefined essential entities. For example, in news summarization the proper nouns are the critical information to retain.

### 4.3 Implementation Details

We implemented our baselines with OpenNMT framework (Klein et al., 2017). We delexicalize utterances according to the belief span annotation. To maintain the generalizability of SPNet, we combine the slots that refer to the same information from different dialog domains into one slot (e.g. time). Instead of using pre-trained word embeddings like GloVe (Pennington et al., 2014), we train word embeddings from scratch with a 128-dimension embedding layer. We set the hidden states of the bidirectional LSTM encoders to 256 dimensions, and the unidirectional LSTM decoder to 512 dimension. Our model is optimized using Adam (Kingma & Ba, 2014) with a learning rate of 0.001, $\beta_1 = 0.9$, $\beta_2 = 0.999$. We reduce the learning rate to half to avoid overfitting when the validation loss increases. We set the hyperparameter $\lambda$ to 0.5 in the objective function and the batch size to eight. We use beam search with a beam size of three during decoding. We use the validation set to select the model parameter. Our model with and without multi-task takes about 15 epochs and seven epochs to converge, respectively.

### 5 Results and Discussions

#### 5.1 Automatic Evaluation Results

To demonstrate SPNet's effectiveness, we compare it with two state-of-the-art methods, Pointer-Generator (See et al., 2017) and Transformer (Vaswani et al., 2017). Pointer-Generator is the state-of-the-art method in abstractive document summarization. In inference, we use length penalty and coverage penalty mentioned in Gehrmann et al. (2018). The hyperparameters in the original implementation (See et al., 2017) were used. Transformer uses attention mechanisms to replace recurrence for sequence transduction. Transformer generalizes well to many sequence-to-sequence problems, so we adapt it to our task, following the implementation in the official OpenNMT-py documentation.

We show all the models' results in Table 1. We observe that SPNet reaches the highest score in both ROUGE and CIC. Both Pointer-Generator and Transformer achieve high ROUGE scores, but a relative low CIC scores. It suggests that the baselines have more room for improvement on preserving critical slot information. All the scaffolds we propose can be applied to different neural network models. In this work we select Pointer-Generator as our base model in SPNet because we observe that Transformer only has a small improvement over Pointer-Generator but is having a higher cost on training time and computing resources. We observe that SPNet outperforms other methods in all the automatic evaluation metrics with a big margin, as it incorporates all the three semantic scaffolds. Semantic slot contributes the most to SPNet's increased performance, bringing the largest increase on all automatic evaluation metrics.

| Conversation |
| --- |
| **Client:** I'm looking for a place to stay. It needs to be a guesthouse and include free wifi. |
| **Help desk:** There are 23 hotels that meet your needs. Would you like to narrow your search by area and/or price range ? |
| **Client:** I would like for it to be cheap and include free parking. |
| **Help desk:** There are nine guesthouse hotels in various areas. What part of town are you hoping for? |
| **Client:** Nothing in particular. I just need it booked for 6 people for a total of 4 nights starting from sunday. I'd also like the reference number, please. |
| **Help desk:** Sure! We will book you in a hotel with 4 stars. The reservation will be for 6 people for 4 nights. How many rooms would you like to reserve? |
| **Client:** Well, for six people I suppose I will be needing three rooms, for four nights, starting Sunday. |
| **Help desk:** You're booked at the Alexander Bed and Breakfast, 517a coldham lane, for 6 people for four nights starting Sunday. Your reference number is JXVKZ7KV. |
| **Client:** Thank you! I'm also looking for a restaurant. Ideally an Italian place in the same price range in the centre. |
| **Help desk:** I found Ask restaurant would you like to book that? |
| **Client:** Is the name of the restaurant Ask restaurant? |
| **Help desk:** Yes, the restaurant is called Ask restaurant. Would you like to book a table there? |
| **Client:** Yes. Book it for the same group of people on the same day at 18:45. Can I get the reference number please? |
| **Help desk:** Great! Your booking for 6 people at the restaurant Ask on Sunday at 18:45 was successful! Your reference number is Y2Y8QYBY. |
| **Client:** I am all set. Thank you. |
| **Help desk:** Thank you for using the Cambridge TowneInfo Help Desk! |

| Summaries |
| --- |
| **Pointer-Generator**: |
| You are traveling to Cambridge and looking forward to try local restaurants. You are looking for a place to stay. The hotel should include free wifi and should be in the type of guesthouse. The hotel does n't need to include internet. If there is no such hotel, how about one that has free wifi. Once you find the hotel you want to book it for 6 people and 4 nights starting from Sunday. If the booking fails how about 1 nights. Make sure you get the reference number. You are also looking for information particular restaurant. Its name is called Ask restaurant restaurant. Make you you the price range and postcode. |
| |
| **SPNet**: |
| You are looking for information in Cambridge. You are looking for a place to stay. The hotel should be in the type of guest house and should include free wifi. The hotel should be in the cheap price range. Once you find the hotel you want to book it for 6 people and 4 nights starting from Sunday. Make sure you get the reference number. You are also looking for a restaurant. The restaurant should be in the same price range as the hotel and should serve Italian food. The restaurant should be in the same area as the hotel. Once you find the restaurant you want to book a table for the same group of people at 18:45 on the same day. Make sure you get the reference number. |
| |
| **Ground truth**: |
| You are planning your trip in Cambridge. You are looking for a place to stay. The hotel should include free wifi and should be in the type of guest house. The hotel should be in the cheap price range and should include free parking. Once you find the hotel you want to book it for 6 people and 4 nights starting from Sunday. Make sure you get the reference number. You are also looking for a restaurant. The restaurant should be in the same price range as the hotel and should be in the centre. The restaurant should serve italian food. Once you find the restaurant you want to book a table for the same group of people at 18:45 on the same day. Make sure you get the reference number. |

Table 2: An example dialog and Pointer-Generator, SPNet and ground truth summaries. We underline semantic slots in the conversation. Red denotes incorrect slot values and green denotes the correct ones.

## 5.2 HUMAN EVALUATION RESULTS

We also perform human evaluation to verify if our method's increased performance on automatic evaluation metrics entails better human perceived quality. We randomly select 100 test samples from MultiWOZ test set for evaluation. We recruit 150 crowd workers from Amazon Mechanical Turk. For each sample, we show the conversation, reference summary, as well as summaries generated by Pointer-Generator and SPNet to three different participants. The participants are asked to score each summary on three indicators: relevance, conciseness and readability on a 1 to 5 scale, and rank the summary pair (tie allowed).

We present human evaluation results in Table 3. In the scoring part, our model outperforms Pointer-Generator in all three evaluation metrics. SPNet scored better than Pointer-Generator on relevance and readability. All generated summaries are relatively concise; therefore, they score very similar in conciseness. Ground truth is still perceived as more relevant and readable than SPNet results. However, ground truth does not get a high absolute score. From the feedback of the evaluators, we found that they think that the ground truth has not covered all the necessary information in the conversation, and the description is not so natural. This motivates us to collect a dialog summarization dataset with high-quality human-written summaries in the future. Results in the ranking evaluation show more differences between different summaries. SPNet outperforms Pointer-Generator with a large margin. Its performance is relatively close to the ground truth summary.

| Summary | Relevance | Conciseness | Readability |
|---|---|---|---|
| Ground truth | 3.83 | 3.67 | 3.87 |
| Pointer-Gen(See et al., 2017) | 3.56 | 3.58 | 3.64 |
| SPNet | **3.77** | **3.67** | **3.80** |
| **Rank Pair** | **Win** | **Lose** | **Tie** |
| SPNet vs. Pointer-Gen | 61.3 | 30.3 | 8.3 |
| SPNet vs. Ground truth | 32.3 | 48.0 | 19.7 |

Table 3: The upper is the scoring part and the lower is the the ranking part. SPNet outperforms Pointer-Generator in all three human evaluation metrics and the differences are significant, with the confidence over 99.5% in student t test. In the ranking part, the percentage of each choice is shown in decimal. Win, lose and tie refer to the state of the former summary in ranking.

### 5.3 CASE STUDY

Table 2 shows an example summary from all models along with ground truth summary. We observe that Pointer-Generator ignores some essential fragments, such as the restaurant booking information (6 people, Sunday, 18:45). Missing information always belongs to the last several domains (restaurant in this case) in a multi-domain dialog. We also observe that separately encoding two speakers reduces repetition and inconsistency. For instance, Pointer-Generator's summary mentions "free wifi" several times and has conflicting requirements on wifi. This is because dialogs has information redundancy, but single-speaker model ignores such dialog property.

Our method has limitations. In the example shown in Table 2, our summary does not mention the hotel name (Alexander Bed and Breakfast) and its address (517a Coldham Lane) referred in the source. It occurs because the ground truth summary doe not cover it in the training data. As a supervised method, SPNet is hard to generate a summary containing additional information beyond the ground truth. However, in some cases, SPNet can also correctly summarize the content not covered in the reference summary (see Table 6 in Appendix).

Furthermore, although our SPNet achieves a much-improved performance, the application of SPNet still needs extra annotations for semantic scaffolds. For a dialog dataset, speaker role scaffold is a natural pattern for modeling. Most multi-domain dialog corpus has the domain annotation. While for texts, for example news, its topic categorization such as sports or entertainment can be used as domain annotation. We find that semantic slot scaffold brings the most significant improvement, but it is seldom explicitly annotated. However, the semantic slot scaffold can be relaxed to any critical entities in the corpus, such as team name in sports news or professional terminology in a technical meeting.

### 6 CONCLUSION AND FUTURE WORK

We adapt a dialog generation dataset, MultiWOZ to an abstractive dialog summarization dataset. We propose SPNet, an end-to-end model that incorporates the speaker role, semantic slot and dialog domain as the semantic scaffolds to improve abstractive summary quality. We also propose an automatic evaluation metric CIC that considers semantic slot relevance to serve as a complementary metric to ROUGE. SPNet outperforms baseline methods in both automatic and human evaluation metrics. It suggests that involving semantic scaffolds efficiently improves abstractive summarization quality in the dialog scene.

Moreover, we can easily extend SPNet to other summarization tasks. We plan to apply semantic slot scaffold to news summarization. Specifically, we can annotate the critical entities such as person names or location names to ensure that they are captured correctly in the generated summary. We also plan to collect a human-human dialog dataset with more diverse human-written summaries.

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

## A   SUPPLEMENT TO CASE STUDY

| Supplement Summary |
|---|
| **Transformer**: You are planning your trip in Cambridge. You are looking for a place to stay. The hotel doesn't need to include internet and should include free parking. The hotel should be in the type of guesthouse. If there is no such hotel, how about one that is in the moderate price range? Once you find the hotel, you want to book it for 6 people and 4 nights starting from Sunday. Make sure you get the reference number. You are also looking forward to dine. The restaurant should be in the centre. Make sure you get the reference number. |

| Human Evaluation |
|---|
| Which one is better in your opinion, summary 1 (Pointer-Generator) or summary 2 (SPNet) or tie? |
| **Evaluator**: Summary 2 (SPNet) |
| What is the reason for your rank? We recommend you to talk about it from relevance, conciseness and readability. |
| **Evaluator**: Summary 2 is more coherent and relevant. It is just as concise but more readable. |
| |
| Which one is better in your opinion, summary 2 (SPNet) or summary 3 (Ground Truth) or tie? |
| **Evaluator**: Tie |
| What is the reason for your rank? We recommend you to talk about it from relevance, conciseness and readability. |
| **Evaluator**: Summary 2 and 3 are almost identical. They are equally coherent, relevant, and readable. |

Table 4: Supplement to the case in Table 2. The summary generated by Transformer is shown in supplement summary. Red denotes incorrect slot values and green denotes the correct ones. Human Evaluation part provides the evaluator's choice and feedback in ranking summary pairs. Content in the brackets is not shown to the evaluators.

## B  DIALOG SUMMARIZATION CASES

| Conversation |
|---|
| **Client**: I'm looking for a moderately priced place to eat that's in the centre of town. |
| **Help desk**: What type of cuisine are you looking for? There are 21 restaurants in that area. |
| **Client**: I don't have a preference. |
| **Help desk**: Out of the 21 restaurant choices, one is the Yippee Noodle Bar which is moderately priced in the centre of town. Would you like to make a reservation? |
| **Client**: That sounds great, what is the postcode? |
| **Help desk**: Their postcode is cb11lh and if you want a reservation just let me know the time, party size, and day and I'll make it for you. |
| **Client:** Not now, thanks. Here's a weird one, though - are there any expensive 0-star hotels in the center of town? |
| **Help desk**: I'm sorry, but there are no expensive 0-star hotels in the area. Can I help you find something else? |
| **Client**: Do you have a hotel in the moderate price range? |
| **Help desk**: The Cityroomz Hotel is moderately priced, and has free internet and parking. Would you like to make a reservation now? |
| **Client**: That sounds nice, yes please book a room for 3 guests staying for 5 nights starting on Sunday. |
| **Help desk**: I'm sorry, those particular dates are filled, if you'd like I could try to make the trip shorter or maybe change days? |
| **Client**: How about just for the one night? |
| **Help desk**: Alright, I was able to book your hotel room at the Cityroomz, for 3 people, for one night. Your reference number is E62IXZGB. Is there anything else I can do? |
| **Client**: I will also need a taxi to go from the restaurant to the hotel, I want to leave the restaurant at 02:15. |
| **Help desk**: I have booked your taxi. It will be a blue volvo and the contact number is 07729446320. |
| **Client**: Great. Thanks. That's all I need today. |
| **Help desk**: You're welcome. Have a wonderful visit. |

| Summaries |
|---|
| **Pointer-Generator**: |
| You are traveling to Cambridge. You are looking for a restaurant. The restaurant should serve 1 food and should be in the moderate price range. Make sure you get postcode and address. You are also looking for information in the centre. The hotel should have a star of 0 and should include free wifi. The hotel doesn't need to include internet. If there is no such hotel, how about one that doesn't have free parking? Once you find the hotel, you want to book it for 3 people and 5 nights starting from Sunday. If the booking fails, how about 1 night? Make sure you get the reference number. |
| |
| **SPNet**: |
| You are traveling to Cambridge and looking forward to try local restaurants. You are looking for a place to dine. The restaurant should be in the moderate price range and should be in the centre. Make sure you get postcode. You are also looking for a place to stay. The hotel should be in the type of hotel and should have 0-star. The hotel should be in the same area as the restaurant and should be in the same price range as the restaurant. Once you find the hotel, you want to book it for 3 people and 5 nights starting from Sunday. If the booking fails, how about 1 night? Make sure you get the reference number. You also want to book a taxi to commute between the 2 places. You want to leave the restaurant by 02:15. Make sure you get contact number and car type. |
| |
| **Ground Truth**: |
| You are traveling to Cambridge and looking forward to try local restaurants. The restaurant should be in the centre and should be in the moderate price range. Make sure you get postcode. You are also looking for a place to stay. The hotel should be in the expensive price range and should have a star of 0. The hotel should be in the same area as the restaurant. If there is no such hotel, how about one that is in the moderate price range? Once you find the hotel you want to book it for 3 people and 5 nights starting from Sunday. If the booking fails how about 1 night. Make sure you get the reference number. You also want to book a taxi to commute between the 2 places. You want to leave the restaurant by 02:15. Make sure you get contact number and car type. |

Table 5: An example dialog and Pointer-Generator, SPNet and ground truth summaries. The dialog spans over three domains: restaurant, hotel and taxi. We underline semantic slots in the conversation. Red denotes incorrect slot values and green denotes the correct ones.

| Conversation |
|---|
| **Client**: I would like a British food restaurant in the centre. |
| **Help desk**: Sure, there are 7 restaurants that meet your needs. 4 are moderate and 3 are expensive. Do you have a preference? |
| **Client**: Only the best for my family .. We'll take the expensive one. Book us a table for 5 at 14:00 on Thursday. |
| **Help desk**: I'm sorry I am having difficulty making a reservation for you. Shall we try another time or restaurant type? |
| **Client**: Let's try Italian instead. |
| **Help desk**: Caffe Uno is a very nice, expensive Italian restaurant in the center of town. Would you like a table there? |
| **Client**: Actually, I change my mind. I think I want to stick with British food after all. Can you suggest any one that's in the centre of town? |
| **Help desk**: Fitzbillies Restaurant, the Cambridge Chop House, and Midsummer House are all centrally located, British, and very expensive, but none have a table for 5 available on Thursday at 14:00. |
| **Client**: Can you try the Cambridge Chop House for Thursday for 5 people at 13:00 instead? |
| **Help desk**: Your reservation at the Cambridge Chop House for 5 people at 13:00 on Thursday was successful. Your reference number is U6GV5ZZV. Is there anything else I can help you with today? |
| **Client**: No, that's all I need. Thanks for your help! |
| **Help desk**: You're welcome, thank you for using our system. |
| **Client**: You're welcome! It was great! |
| **Help desk**: Have a good day! |

| Summaries |
|---|
| **Pointer-Generator**: |
| You are looking for a restaurant. The restaurant should serve British food and should be in the centre. Once you find the restaurant you want to book a table for 5 people at 14:00 on Thursday. Make sure you get the reference number. |
| |
| **SPNet**: |
| You are looking for a place to dine. The restaurant should serve British food and should be in the centre. The restaurant should be in the expensive price range. If there is no such restaurant, how about one that serves British food? Once you find the restaurant you want to book a table for 5 people at 13:00 on Thursday. If the booking fails, how about different restaurant in the same area and price range? Make sure you get the reference number. |
| |
| **Ground Truth**: |
| You are looking for a restaurant. The restaurant should serve British food and should be in the centre. Once you find the restaurant, you want to book a table for 5 people at 14:00 on Thursday. If the booking fails, how about 13:00? Make sure you get the reference number. |

Table 6: An example dialog and Pointer-Generator, SPNet and ground truth summaries. The dialog spans over one domain: restaurant. We underline semantic slots in the conversation. Red denotes incorrect slot values and green denotes the correct ones. Blue denotes the content not covered by ground truth in SPNet's summary.

