# OpenReview forum: "Abstractive Dialog Summarization with Semantic Scaffolds"
_ICLR.cc/2020/Conference — Reject_

### Official Review · AnonReviewer1 · 2019-10-19
**Official Blind Review #1**

**Rating:** 1

**Review:**

The paper "Abstractive Dialog Summarization with Semantic Scaffolds" presents a new architecture that the authors claim is more suited for summarizing dialogues. The dataset for summarization was synthesized from an existing conversation dataset.

The new architecture is a minor variation of an existing pointer generator network presented by See et al. First the authors used two different sets of parameters to encode the user and the system responses. The authors also pre-process the dialog by replacing the slot values by their slot keys. Finally, the authors use an auxiliary task of detecting the domain of the dialog.

These three different enhancements are all called "scaffolds" by the authors, hance the title of the paper.

This paper is not suited for ICLR because of its limited novelty. The three enhancements proposed by the authors are long known and incremental.

**Experience Assessment:**

I have published one or two papers in this area.

**Review Assessment: Checking Correctness Of Derivations And Theory:**

N/A

**Review Assessment: Checking Correctness Of Experiments:**

I carefully checked the experiments.

**Review Assessment: Thoroughness In Paper Reading:**

I read the paper thoroughly.

---

### Official Review · AnonReviewer3 · 2019-10-20
**Official Blind Review #3**

**Rating:** 1

**Review:**

Authors proposed an enhanced Pointer-Generator model called SPNet. The key difference between SPNet and PG are the separate handling or using of speaker role, semantic slot and domain labels. Authors also proposed a new metrics called Critical Information Completeness (CIC) to address ROUGE's weakness in assessing if key information is missing in the output.

SPNet considers speak role by using separate encoders for each speaker in the dialog. The hidden state vectors of all speakers are concatenated for next layer.

Semantic slot is modeled by delexicalizing the input, i.e. replacing values (18:00) with their semantic category (time). The actual value is later recovered from input text by copying over the corresponding raw tokens according to the attention layer. The domain labels are incorporated by combining categorization task loss into the final training loss.

Authors used the MultiWoz dataset to evaluate the model and compared it with state-of-the-art Pointer-Generator and Transformer models. ROUGE and proposed CIC metrics all show clear improvements in SPNet. The best performance was observed when all three improvements over SPNet are leveraged. Authors also provided example generated summary and discussed the difference between SPNet PG and baseline. An additional human evaluation was conducted which confirmed the quality gain.

The main concern of Reviewer is the inconsistency in the paper.

1) Authors claimed to "propose an abstractive dialog summarization dataset based on MultiWOZ (Budzianowski et al., 2018)" in the abstract and introduction, which sounds like part of their contribution is creating a new dataset, but in experiment section there's no discussion about how the dataset was created or used at all. The same claim reappeared as the first sentence in the conclusion section.

2) Authors emphasized two drawbacks in the beginning of the paper, but didn't discuss or show any evidence of those drawbacks from data later.

The above inconsistency suggests the paper may not be quite ready for publication.

Other issues found by Reviewer:

1) In equation (7), value() seems to be the word while on the right hand side it's a numerical value (max a_i^t). Did Authors mean argmax?

2) In Table 1, dialog domain seems to provide very marginal improvement, does it justify the complexity added?

3) In Section 4.3, why do we need to train a customized embedding? The process and parameter for the embedding training was not described.

4) In Section 4.3 "batch size to eight" better be consistent as "batch size to 8" (minor issue).



**Experience Assessment:**

I have read many papers in this area.

**Review Assessment: Checking Correctness Of Derivations And Theory:**

I carefully checked the derivations and theory.

**Review Assessment: Checking Correctness Of Experiments:**

I carefully checked the experiments.

**Review Assessment: Thoroughness In Paper Reading:**

I read the paper thoroughly.

---

### Official Review · AnonReviewer2 · 2019-10-25
**Official Blind Review #2**

**Rating:** 3

**Review:**

=== Summary ===

The authors propose a new abstractive dialog summarization dataset and task based on the MultiWOZ dataset. Unlike previous work which targets very short descriptions of dialog transcripts (e.g. 'industrial designer presentation'), this paper looks to generate long descriptions of the entire dialog using the prompts in the MultiWOZ task. The authors also extend the pointer generator network of See et al.  (2018) to use speaker, semantic slot and domain information.  They show that this new model (SPNet) outperforms the baseline on existing automatic metrics, on a new metric tuned to measure recall on slots (dubbed CIC), and a thorough human evaluation.

=== Decision ===

The task of abstractive dialog summarization is well motivated and the field sorely needs new datasets to make progress on this task.  This paper is well written and executed, but unfortunately, I lean towards rejecting this paper because of a fundamental flaw in the nature of the proposed dataset that limits its applicability to the task of abstractive dialog summarization (more below).

My key concern is that the references in the dataset are generated from a small number of templates (Budzianowski et. al ,2018), which suggests this task is mostly one of slot detection and less about summarization.  The significant impact of including semantic slot information seems to be strong evidence this is the case. It is possible to rebut this concern with an analysis of how the generated summaries differ from the reference summaries. For example, Table 2 shows that sometimes the ordering of arguments is swapped: how often does this sort of behavior occur and how often do models identify information not in the reference?


**Experience Assessment:**

I have published in this field for several years.

**Review Assessment: Checking Correctness Of Derivations And Theory:**

N/A

**Review Assessment: Checking Correctness Of Experiments:**

I carefully checked the experiments.

**Review Assessment: Thoroughness In Paper Reading:**

I read the paper at least twice and used my best judgement in assessing the paper.

---

### Decision · Program_Chairs · 2019-12-19

**Decision:**

Reject

**Comment:**

This paper proposes an approach for abstractive summarization of multi-domain dialogs, called SPNet, that incrementally builds on previous approaches such as pointer-generator networks. SPNet also separately includes speaker role, slot and domain labels, and is evaluated against a new metric, Critical Information Completeness (CIC), to tackle issues with ROUGE. The reviewers suggested a set of issues, including the meaningfulness of the task, incremental nature of the work and lack of novelty, and consistency issues in the write up. Unfortunately authors did not respond to the reviewer comments. I suggest rejecting the paper.